# Time for Revival of Bone Biopsy with Histomorphometric Analysis in Chronic Kidney Disease (CKD): Moving from Skepticism to Pragmatism

**DOI:** 10.3390/nu14091742

**Published:** 2022-04-22

**Authors:** Maria Fusaro, Giulia Vanessa Re Sartò, Maurizio Gallieni, Laura Cosmai, Piergiorgio Messa, Maurizio Rossini, Iacopo Chiodini, Mario Plebani, Pieter Evenepoel, Nicholas Harvey, Serge Ferrari, Jorge Cannata-Andía, Andrea Trombetti, Maria Luisa Brandi, Markus Ketteler, Thomas L. Nickolas, John Cunningham, Syazrah Salam, Carlo Della Rocca, Aldo Scarpa, Salvatore Minisola, Fabio Malberti, Filomena Cetani, Mario Cozzolino, Sandro Mazzaferro, Luigi Morrone, Giovanni Tripepi, Martina Zaninotto, Maria Cristina Mereu, Maura Ravera, Giuseppe Cianciolo, Gaetano La Manna, Andrea Aghi, Sandro Giannini, Luca Dalle Carbonare

**Affiliations:** 1National Research Council (CNR), Institute of Clinical Physiology (IFC), 56124 Pisa, Italy; 2Department of Medicine, University of Padova, 35128 Padova, Italy; 3Post-Graduate School of Specialization in Nephrology, University of Milano, 20157 Milano, Italy; giulia.resarto@unimi.it (G.V.R.S.); maurizio.gallieni@unimi.it (M.G.); 4Department of Biomedical and Clinical Sciences, Università di Milano, 20157 Milano, Italy; 5Nephrology Unit, ASST Fatebenefratelli Sacco, 20157 Milano, Italy; lacos@iol.it; 6Unit of Nephrology, Dialysis and Kidney Transplantation, Fondazione IRCCS Ca’ Granda Ospedale Maggiore Policlinico di Milano, 20157 Milano, Italy; piergiorgio.messa@policlinico.mi.it; 7Department of Clinical Sciences and Community Health, University of Milano, 20122 Milano, Italy; 8Rheumatology Unit, University of Verona, 37129 Verona, Italy; maurizio.rossini@univr.it; 9Unit for Bone Metabolism Diseases and Diabetes & Lab of Endocrine and Metabolic Research, Istituto Auxologico Italiano, IRCCS, 20157 Milano, Italy; iacopo.chiodini@unimi.it; 10Laboratory Medicine Unit, Department of Medicine, University of Padua, 35121 Padua, Italy; mario.plebani@unipd.it (M.P.); martina.zaninotto@aopd.veneto.it (M.Z.); 11Laboratory of Nephrology, Department of Immunology and Microbiology, KU Leuven, B-3000 Leuven, Belgium; pieter.evenepoel@uzleuven.be; 12MRC Lifecourse Epidemiology Centre, University of Southampton, Southampton SO16 6YD, UK; nch@mrc.soton.ac.uk; 13NIHR Southampton Biomedical Research Centre, University of Southampton and University Hospital Southampton NHS Foundation Trust, Southampton SO16 6YD, UK; 14Service des Maladies Osseuses, Département de Médecine, HUG, 1205 Geneva, Switzerland; serge.ferrari@unige.ch; 15Bone and Mineral Research Unit, Instituto de Investigación Sanitaria del Principado de Asturias (ISPA), REDinREN del ISCIII, Hospital Universitario Central de Asturias, Universidad de Oviedo, 33003 Oviedo, Spain; cannata@hca.es; 16Division of Bone Diseases, Department of Medicine, Geneva University Hospitals and Faculty of Medicine, Rue Gabrielle-Perret-Gentil 4, 1205 Geneva, Switzerland; andrea.trombetti@hcuge.ch; 17Department of Surgery and Translational Medicine, University of Florence, Viale Pieraccini 6, 50139 Florence, Italy; marialuisa.brandi@unifi.it; 18Department of General Internal Medicine and Nephrology, Robert-Bosch-Krankenhaus, 70376 Stuttgart, Germany; markus.ketteler@rbk.de; 19Division of Nephrology, Columbia University Irving Medical Center, New York, NY 10027, USA; tln2001@cumc.columbia.edu; 20Centre for Nephrology, The Royal Free Hospital and UCL Medical School, London WC1E 6BT, UK; drjohncunningham@gmail.com; 21Sheffield Kidney Institute, Sheffield Teaching Hospitals National Health Service Foundation Trust, Sheffield S10 2JF, UK; syazrah.salam@sth.nhs.uk; 22Department of Medico-Surgical Sciences and Biotechnology, Sapienza University, Polo Pontino, 00185 Rome, Italy; carlo.dellarocca@uniroma1.it; 23ARC-Net Centre for Applied Research on Cancer, University and Hospital Trust of Verona, 37134 Verona, Italy; aldo.scarpa@univr.it; 24Department of Pathology and Diagnostics, University and Hospital Trust of Verona, 37134 Verona, Italy; 25Department of Clinical, Internal, Anaesthesiological and Cardiovascular Sciences, Sapienza University of Rome, 00185 Rome, Italy; salvatore.minisola@uniroma1.it; 26UO Nefrologia e Dialisi ASST Cremona, 26100 Cremona, Italy; f.malberti@asst-cremona.it; 27Unit of Endocrinology, Department of Clinical and Experimental Medicine, University of Pisa, 56126 Pisa, Italy; cetani@endoc.med.unipi.it; 28Department of Health Sciences, Renal Division, San Paolo Hospital, University of Milan, 20142 Milan, Italy; mario.cozzolino@unimi.it; 29Nephrologic Unit, Department of Translational and Precision Medicine, University of Rome ‘La Sapienza’, 00185 Rome, Italy; sandro.mazzaferro@uniroma1.it; 30Nephrology, Dialysis and Renal Transplantation Unit, University Hospital “Policlinico”, 70124 Bari, Italy; lfmorrone@gmail.com; 31CNR-IFC, Clinical Epidemiology of Renal Diseases and Hypertension, Ospedali Riuniti, 89124 Reggio Calabria, Italy; gtripepi@ifc.cnr.it; 32Independent Researcher, 09100 Cagliari, Italy; crissmer5412@gmail.com; 33Policlinico San Martino, 16132 Genova, Italy; maura.ravera@hsanmartino.it; 34Nephrology, Dialysis and Renal Transplant Unit, IRCCS Azienda Ospedaliero-Universitaria di Bologna, Alma Mater Studiorum University of Bologna, 40126 Bologna, Italy; giuseppe.cianciolo@aosp.bo.it (G.C.); gaetano.lamanna@unibo.it (G.L.M.); 35Department of Medicine, Clinica Medica 1, University of Padua, 35128 Padua, Italy; andrea.aghi@gmail.com (A.A.); sandro.giannini@unipd.it (S.G.); 36Section of Internal Medicine, Department of Medicine, University of Verona, 37134 Verona, Italy; luca.dallecarbonare@univr.it

**Keywords:** CKD-MBD, chronic kidney disease, renal osteodystrophy, osteoporosis, bone biopsy, fractures

## Abstract

Bone Biopsy (BB) with histomorphometric analysis still represents the gold standard for the diagnosis and classification of different forms of renal osteodystrophy. Bone biopsy is the only technique able to provide comprehensive information on all bone parameters, measuring static and dynamic parameters of turnover, cortical and trabecular microarchitecture, and mineralization defects. In nephrological practice, bone biopsy yields relevant indications to support therapeutic choices in CKD, heavily impacting the management and prognosis of uremic patients. Unfortunately, the use of bone biopsy has decreased; a lack of expertise in performing and interpreting, perceived procedure invasiveness and pain, and reimbursement issues have all contributed to this decline. Nevertheless, both bone biomarkers and instrumental images cannot be considered reliable surrogates for histological findings, being insufficiently accurate to properly evaluate underlying mineral and bone disorders. This is a multidisciplinary position paper from the Nephrology and Osteoporosis Italian Scientific Societies with the purpose of restating the role of bone biopsy in CKD patient management and of providing strong solutions to allow diffusion of this technique in Italy, but potentially also in other countries. The Italian approach through the optimization and standardization of bone biopsy procedure, the construction of the Italian Hub and Spoke network, and a request for adjustment and national homogenization of reimbursement to the Italian Health Ministry has led the way to implement bone biopsy and to improve CKD patient management and prognosis.

## 1. Introduction

Chronic kidney disease (CKD) is a significant public health issue, affecting about 11% of the world’s population [1]. Mineral and bone disorders are complications of CKD that can occur early in the course of the disease, becoming constantly present in patients at advanced stages [1,2]. The definition of CKD mineral and bone disorders (CKD-MBD) syndrome was first coined in 2006 by experts, in the context of Kidney Disease Improving Global Outcomes (KDIGO) guidelines, describing a complex systemic condition that includes one or more of the following: (a) laboratory abnormalities of bone and mineral metabolism involving calcium, phosphorus, parathyroid hormone (PTH), or vitamin D; (b) abnormalities in bone turnover, mineralization, volume, or strength; (c) extra-skeletal calcifications, such as vascular or other soft tissue [3]. It is clear that the CKD-MBD syndrome determines important consequences in terms of fractures (four times higher in end-stage renal disease), cardiovascular events and higher mortality; thus, the pathophysiology of renal bone disease extends beyond the skeleton, needing a broader definition that recognizes the strong link between abnormal bone remodeling activity and risk for vascular calcifications [1].

This new concept emphasizes that renal osteodystrophy (ROD) is one component of the CKD-MDB syndrome, and the term should be limited to specific histological patterns associated with moderate to end-stage CKD [2]. The correct diagnosis and characterization of different ROD forms are essential to guide prevention and therapeutic strategies, with the latter strongly influenced by the type of bone abnormalities. A comprehensive management of CKD-MBD is a relevant challenge in our clinical practice, also considering the limitations of bone biomarkers and feasible bone imaging methods. In this complex scenario, bone biopsy (BB) with histomorphometric analysis represents the gold standard for diagnosis and classification of ROD, providing information that is not available by any other diagnostic tool. Indeed, a meticulous definition of ROD through histomorphometric criteria heavily impacts management and prognosis of uremic patients.

Unfortunately, during the last few decades, the number of BB has decreased because of perceived invasiveness, potential pain associated, lack of expertise in performing and interpreting, and reimbursement issues such that only few centers today still perform this analysis. The aims of this position paper are to assess clinical indications and new applications of BB in CKD and to provide suggestions for its diffusion in Italy and potentially spread them to other countries as well. Furthermore, the formation of a collaborative Italian Network based on Hub and Spoke model should allow improved management of ROD and facilitate clinical research.

## 2. Renal Osteodystrophy (ROD)

**Definition.** Nowadays ROD is a classic term that should be exclusively used to define histological bone lesions related to CKD [2]; this spectrum of abnormalities could be assessed only by BB according to the TMV (turnover, mineralization, volume) classification system, recommended by KDIGO in 2006 to standardize the changes [3,4].

**TMV classification system.** Bone turnover reflects the rate of skeletal remodeling, which is related to the process of bone resorption and formation. In CKD, it varies widely and can be low, normal, or high, and the extremes have been associated with vascular calcifications and higher mortality. Despite the identification of new hormones and markers of bone formation/resorption (e.g., bone alkaline phosphatase, tartrate-resistant acid phosphatase 5b, osteocalcin, sclerostin, klotho and fibroblast growth factor 23), diagnostic accuracy has been difficult to establish in CKD patients [4]. In contrast, turnover assessed with histomorphometry by significant increase or reduction in osteoblast and osteoclast numbers and by dynamic measurements of osteoblast function, using the tetracycline double-labeled technique, is accepted as a robust measurement [2,4].

Mineralization reflects the physiological share of collagen that becomes calcified during bone formation. In CKD, it can be normal or deficient. Routine serum biomarkers (e.g., calcium, phosphorus, and total alkaline phosphatase) are not specific or sensitive enough to identify mineralization disorders; it is assessed with histomorphometry both by static measurements of osteoid (volume, thickness) and by dynamic tetracycline-based measurements of mineralization lag time [2].

Volume reflects the amount of bone per unit of tissue. In CKD it can be low, normal, or high and various imaging tools could help (i.e., dual-energy X-ray absorptiometry, high resolution peripheral quantitative computerized tomography, or trabecular bone score software). In clinical practice, none of these methods allow evaluation of turnover and mineralization and there are few data on CKD patients. Volume is assessed with histomorphometry by static measurements [2].

**Histological classification**. The different combinations of each of these histological descriptors characterize specific entities in CKD. The classic description includes secondary hyperparathyroidism indicated as osteitis fibrosa, adynamic bone disease, osteomalacia, and mixed uremic osteodystrophy. The TMV classification system is shown in Table 1. This standardized nomenclature for patterns is needed to promote a more widespread understanding and to facilitate comparisons from various research reports [1,2,3]. Suggestive representations of these different forms of ROD are obtained by BB approach, allowing the study of each component of the bone multicellular unit (Figure 1) [5].

**Uremic osteoporosis and bone quality.** Advanced stages of CKD are characterized by skeletal fragility, with a risk of fractures four to six-fold higher than age and gender-matched controls without CKD [6]. In these patients impaired bone strength and bone quality assert a certain relation between CKD and osteoporosis.

The pathophysiology is complex, consisting of a mixture of classic osteoporosis, drug-induced bone disorders, and CKD-MBD syndrome [6]. Firstly, primary osteoporosis (age and sex-related or due to other classic factors) could play a more prominent role in CKD bone fragility than previously recognized, impacting early in the course of the disease. Second, a multitude of drugs may cause negative bone effects (e.g., steroids, loop diuretics, heparin, proton pump inhibitors or vitamin K antagonists) but also poor levels of nutrients and vitamins (e.g., vitamin D and vitamin K) are recently recognized to be associated with osteoporosis and fracture risk [7,8]. Third, in CKD patients, bone strength and quality are impaired by the accumulation of uremic toxins, such as indoxyl sulfate, *p*-cresyl sulfate, and advanced glycation end products (AGE), that create oxidative stress causing impairment, especially in material properties [4,9]. Bone quality is the combination of both structural properties, such as cortical and trabecular microarchitecture, and material properties, such as mineral composition and collagen type I crosslinking, enabling bone to tolerate load and resist fractures. In uremic bone, there is a prevalence of pathological collagen cross-links and immature crystals [9,10]. Moreover, Hsu et al., highlighted how the uremic environment affects bone quality, particularly through reduced expression of the PTH receptor in osteoblasts and association with skeletal resistance to PTH [11].

Thus, a new concept of ‘uremic’ osteoporosis, characterized by low volume but normal turnover and mineralization, is introduced emphasizing the strict relationship between end-stage renal disease (dialysis) and an increase in bone fractures, presuming a potential role into preventing them through the use of dialytic methods with high removal of uremic toxins (e.g., high flux membrane hemodialysis, hemodiafiltration) [10] (Table 1, Figure 2). BB analysis in future studies and clinical practice should investigate this issue with respect of fracture risks.

## 3. ROD Changes over Decades

The epidemiology of ROD has changed for reasons not completely clear, but in part as a consequence of new drugs in the management of MBD and increased survival of patients. High turnover bone disease has long been the predominant type of ROD, while in the last two decades, low turnover disorders have become prominent [12,13]. In particular, Malluche et al., observed this change in 630 bone biopsies of CKD stage 5 patients on maintenance dialysis, performed in Europe or the USA from 2003 to 2008. Distancing from the results of older studies, 58% of patients had a low bone turnover disease. Interestingly, there were racial differences with white patients exhibiting predominantly low turnover (62%), whereas black patients showed mostly normal or high turnover (68%) [12]. Recently, data collected in the Brazilian Registry of Bone Biopsy (REBRABO), which includes BB data of 260 CKD stage 3–5 patients, have not confirmed the trend [14]. Osteitis fibrosa, mixed uremic osteodystrophy, adynamic bone and osteomalacia have been detected in 50%, 25%, 16% and 6% of patients, respectively. In the same cohort, a high prevalence of osteoporosis and aluminium accumulation was also detected.

In the current literature, the actual prevalence of different types of ROD is still debated and this could be related to many factors including epidemiological and clinical differences, such as age and/or ethnicity among CKD patients. Furthermore, differences between health systems could influence the availability of drugs and therapeutic strategies, leading to different types of bone disorders. Histomorphometric data related to CKD patients not on dialysis and to renal transplant recipients are, then, sparse, but some findings show a trend to lower bone turnover for both categories [15,16].

## 4. Why We Need to Perform Bone Biopsy in CKD Patients

Management of mineral and bone disorders and control of fracture burden in CKD patients is a great challenge, requiring first the determination of whether the individual patient has osteoporosis and/or various forms of ROD to enable the choice of the optimal therapeutic strategy [17]. Histomorphometric analysis by BB is a candidate to be a reliable and reproducible diagnostic tool to support adequate therapies. On the other hand, difficulties and limitations to perform BB have motivated nephrologists to search for non-invasive alternatives such as bone biomarkers and/or instrumental images to validate a correct surrogate to histological findings.

**Bone biomarkers and BB.** Circulating biomarkers commonly used to discriminate between high and low bone turnover in clinical practice are PTH and alkaline phosphatase (ALP).

PTH has been for a long time considered a pivotal marker in the management of CKD as a surrogate of bone turnover guiding therapeutic decisions, but recently its role has been reconsidered as a consequence of inconsistent results in various studies. Indeed, a histomorphometric study on bone turnover (492 dialysis patients with prior BB) showed concentrations of PTH > 323.0 pg/mL have the best discriminatory ability between high and normal/low bone turnover, and of PTH < 103.8 pg/mL best predicts low from non-low turnover [13]. In another study, conducted on 97 hemodialysis patients, PTH levels were compared with bone histological findings both at the onset of the study and after completing 1 year follow up; low and high bone turnover patterns were the most prevalent disorders identified below the lower and above the upper limit of K/DOQI PTH range (150–300 pg/mL), respectively; however, while the lower cut-off level of PTH (<150 pg/mL) provided a positive predictive value of 83% for identifying low bone turnover, PTH level greater than 300 pg/mL was not equally adequate for the diagnosis of high turnover (positive predictive value 62%); in fact, even for upper cut-off level of PTH range, a histomorphometric pattern of low turnover was found in one-third of the patients assessed by BB. Notably, in the group targeting the recommended range of PTH, low bone turnover was the most common finding (64%), not guaranteeing the prevention of bone remodeling in advanced CKD patients [18]. PTH shows poor predictive power in measuring bone turnover and, unless at the extremes, performs poorly as a bone biomarker.

ALP was largely abandoned from the mid-1990s because of the ready availability of PTH assays; however, recent prospective studies have reassessed the role of ALP as a marker of fracture risk in advanced CKD and hemodialysis patients. A large Japanese cohort study of 185,277 hemodialysis patients (median hemodialysis duration of 5.8 years), assessed 1 year follow up biochemical parameters and clinical outcomes; hip fractures were newly diagnosed in 1586 patients (1.0%) and 14,230 of them (7.9%) died of all causes, including from cardiovascular issues. The results highlighted a significant association between increased serum ALP with a higher incidence of hip fractures and higher mortality: patients in the highest ALP quartile had higher risk of both all cause and cardiovascular mortalities than those in the lowest one (OR 1.46, 95% CI 1.33–1.60 and OR 1.25, 95% CI 1.10–1.42, respectively); patients in the in the highest ALP quartile had also a higher incidence of hip fractures (OR 1.71, 95% CI 1.33–2.18), especially in patients with lower PTH concentrations [19]. Then, Imori et al., in a 5-year prospective study on 485 patients on hemodialysis, studied fracture risk associated with serum biomarkers; in the occurrence of a rate of 1.9 new fractures per 100 patient-years, a higher level of ALP was stated as a significant predictor of any type of incident fracture (area under curve 0.766, *p* < 0.0001). The stratified analysis by different PTH levels also demonstrated a greater risk of any type of fractures associated with either low (<150 pg/mL, hazard ratio 3.47, *p* < 0.01) or high PTH levels (>300 pg/mL, HR 5.88, *p* < 0.0001) [20].

Additional turnover biomarkers have been identified, such as cross-linked collagen type I peptide (CTX), tartrate-resistant acid phosphatase 5b (TRAP-5b) for bone resorption and procollagen type 1 *n*-terminal propeptide (P1NP) for bone formation. A recent study tested the diagnostic accuracy of biomarkers such as ALP, intact P1NP, TRAP-5b and of high resolution peripheral quantitative computerized tomography (HR-pQCT) in predicting bone turnover status in CKD, compared to histomorphometric results of BB. The results of this study, conducted on 69 CKD stage 4–5 and 68 control participants (total 43 evaluable BB), demonstrated the ability of both biomarkers and HR-pQCT to discriminate low from non-low bone turnover in advanced CKD (in ROC analysis, all their AUCs are >0.79) [21]. Nevertheless, in clinical practice, their availability is still limited, and their accuracy still needs to be better assessed and standardized.

**Instrumental images and BB.** Several non-invasive imaging methods are available to investigate bone volume and characterize the CKD-MBD syndrome.

Dual-energy X-ray absorptiometry (DXA) investigates bone density, a measure of mineralized bone mass, and a predictor of fragility fractures. Although the updated KDIGO 2017 guidelines have highlighted the relevance of DXA in CKD stage 3a–5 patients in assessing fracture risk, this method is unable to specify ROD types and thus to give indications on therapy for the underlying bone disease. DXA does not discriminate between cortical and trabecular bone density, neither it is able to evaluate microarchitecture, bone mineralization or turnover. These major issues limit its use in CKD patients, despite wide availability [2,9].

HR-pQCT allows a three-dimensional assessment of bone density and an evaluation of cortical and trabecular bone quality (microarchitecture and material properties) of entire bone but also separate compartments. Then, Finite Element Analysis of the images obtained allows for simulating the ability to resist an impact force with a mathematical approach. In CKD patients, HR-pQCT was considered a non-invasive tool to study bone disorders and the cortical measures may represent a new perspective on the fracture risk assessment in dialysis patients; indeed, a study conducted in 52 hemodialysis patients (more than half with a history of fracture), observed an important reduction of cortical parameters (i.e., cortical bone density, area, and thickness), investigated by radial pQCT [22]. In another study, Nickolas et al., demonstrated, with the use of HR-pQCT in 55 patients with CKD stage 2–5, a significant deterioration in the cortical area (−2.9%), bone density (−1.3%), thickness (−2.8%), and an increase in its porosity (+4.2%) [23]. Interestingly, a comparison between BB and HR-pQCT, including 31 dialysis patients, found a statistically significant agreement between histomorphometric and HR-pQCT microstructural parameters: trabecular density and bone volume/total volume (BV/TV), measured by HR-pQCT, predicted trabecular bone volume assessed by BB; also, trabecular number, separation, and thickness correlated between the two methods. In this study, patients with cortical porosity by BB presented lower cortical density at the distal radius [24]; however, from current evidence, the role of HR-pQCT is still debated and does not give information on turnover and mineralization, limiting its use in clinical practice to evaluate CKD-MBD and specific settings.

^18^F-Sodium Fluoride positron emission tomography (^18^F-NaF PET) is a valuable dynamic imaging technique that measures regional bone turnover at clinically relevant sites (e.g., lumbar spine and anterior iliac crest). The tracer reflects the metabolic activity and remodeling in the bone. A significant correlation between fluoride activity in the PET scan and dynamic histomorphometric parameters by BB has been clearly established in 26 dialysis patients—according to the reference values by Malluche, ^18^F-NaF PET discriminates low from non-low turnover with a sensitivity of 76% and specificity of 78% [25]. The subsequent analysis compared results with the unified TMV-based classification system, demonstrating improvement in diagnostic accuracy of ^18^F-NaF PET to differentiate types of ROD (AUC 0.86 and 0.87 for discriminating high turnover or low turnover bone disease, respectively, from other types of ROD) [26]. Despite the promising data, ^18^F-NaF PET measures turnover and remodeling and cannot distinguish mineralizing defects: more research is needed to adapt results in clinical practice and to establish the role of this imaging tool in CKD settings.

Complementary information on bone status can be obtained with microindentation, a minimally invasive technique able to measure micromechanical bone properties with parameters related to stiffness, material strength, and tissue elements (collagen maturity, non-collagen proteins, osteon characteristics, hydration, microdamage and microporosity) [27]. The test involves pressing a hard tip, with a known force, into the specific bone region of interest, measuring the hardness of the contact area on a microscopic scale; the deeper is the indentation, the less resistant is bone. Reference Point Indentation (RPI) technique, with the use of a handheld device (OsteoProbe or BioDent), is recently developed to measure most likely subperiosteal bone material properties, such as indentation distance and bone material strength index (BMSi), providing an important contribution in understanding the pathogenesis of bone fragility.

These measures have the potential in CKD patients to improve the estimation of fracture risk. The first studies in humans using RPI demonstrated deterioration of bone properties in small cohorts of patients with hip and atypical femoral fractures. Additional studies have explored BMSi measures in different populations; in a cohort of 53 kidney transplanted patients, an important deterioration in all markers of bone density and strength, the latter assessed by microindentation device, has been demonstrated compared to 94 controls (BMSi 79, 95% CI 71.8–84.2 versus 82, 95% CI 77.5–88.9, adjusted *p* = 0.005), making microindentation a potential tool for identifying high-risk patients of fractures after transplantation [28]. Of interest, BMSi showed no correlation with CKD status and performed similarly in identifying fracture probability, regardless of renal disease [29].

To date, the microindentation data, available from the literature, are still unclear and its role in prediction of fractures is not fully established. Further research, also including CKD patients, is needed to compare this technology with other mechanical testing methods and to correctly determine its power to identify high fracture risk patients [27,30].

As a whole, both bone biomarkers and instrumental images have not resulted in sufficiently accurate data to evaluate optimal bone turnover, volume and mineralization to prevent fragility fractures. To date, BB remains the gold standard for diagnosis and specific classification of ROD in CKD patients, providing comprehensive information on all bone parameters. Performing this analysis before starting any therapy, is key to give relevant indications to nephrologists to support CKD-MBD therapeutical choices in short and long term [2,31].

Unfortunately, the number of BB has significantly declined in the last years, both for clinical and research purposes, as confirmed by a 2015 European survey [32]; many limitations, collected by this recent survey, such as the availability of technical, clinical, and pathological expertise, costs, or inadequate reimbursement, are continuing to threaten its use. This decline also reflects the 2017 revised KDIGO guidelines suggestion to perform BB in CKD stage 3–5 patients only if the type of ROD will impact treatment decisions [33]. This is, to a certain extent, far from 2009 guidelines that strongly recommended to use BB in at least five key situations (multiple fractures, unexplained hypercalcemia, or hypophosphatemia, suspected aluminium toxicity, or before osteoporosis treatment) [34]. The latest Canadian Society of Nephrology recommendations surprisingly discourages routine BB in clinical practice, enhancing the current difficulties to perform it [35].

In our opinion, the positive impact of BB in management of CKD-MBD is strongly significant to be discouraged. For this reason, our aim is to propose solutions to current limitations to allow wider uptake of this technique.

## 5. Indications of Bone Biopsy

BB provides quantitative histological examination aimed at obtaining information on bone remodeling, structure, and microarchitecture. It is the only technique able to assess both cortical and trabecular microarchitecture; furthermore, it evaluates static and dynamic parameters of osteoblastic activity and the mineralization process, through double tetracycline labeling before the procedure [36]. BB currently remains the gold standard to confirm diagnosis of changes in mineral and bone metabolism and characterize different forms of ROD, providing most relevant information to guide choices in CKD-MBD management. Thus, the therapeutic approach in CKD-MBD is challenging and complicated by the fact that CKD patients may be affected both by aspects related to osteoporosis and by specific changes due to underlying forms of ROD. Performing BB before starting new treatment of CKD-MBD, allows to choose the best feasible strategies according to bone turnover: in the case of adynamic bone disease, the use of antiresorptive drugs (e.g., bisphosphonates, denosumab) is avoided preventing the excessive suppression of skeletal remodeling and/or precipitation of a condition of low turnover. Conversely, in the case of ROD with high turnover, the use of anabolic drugs (e.g., teriparatide, and not yet prescribed abaloparatide and romosozumab), is restricted limiting the significant increase in skeletal turnover. BB could be useful even as a follow-up technique, able to verify the therapeutic effect of antiresorptive or anabolic drugs or their potential side effects on bone [37,38].

The indications to perform BB include several situations: 2009 KDIGO CKD-MBD guidelines recommended the need for a BB in at least five important circumstances (e.g., multiple fractures, unexplained hypercalcemia, or hypophosphatemia, suspected aluminium toxicity, or before osteoporosis treatment); however, this statement has been expanded by the 2017 update, which suggests performing BB in CKD stage 3–5 patients if the type of ROD will impact treatment decisions [33,34]. Indications have also been described by recent publications, such as the European survey of 2015 and The Brazilian Registry of Bone Biopsy (REBRABO) [14,32], providing real knowledge of the most frequent reasons that led to performing BB in clinical practice. A comparison of the main indications, collected from literature, is summarized in Table 2.

## 6. Technique and Potential Combinations of Bone Biopsy for an Innovative Approach

The procedure establishes (Figure 3), before performance of biopsy, the tetracycline double-labeled technique as the best approach to investigate dynamic parameters of bone turnover, bone formation rates, and mineralization defects. Tetracycline compounds chelate calcium on bone surfaces, and they are deposited inside the bone at sites of active mineralization. The ‘labels’ could be visualized in fluorescent light, during histomorphometric evaluation. Tetracycline hydrochloride 500 mg o.d., twice daily, is prescribed for 2–3 days and then on day 15 and day 16, after an intermission of 12 days. The BB is scheduled in accordance with this prescription, after 5–7 days from the last administration [36]. Double bands of tetracycline fluorescence can be seen circumscribing the amount of new bone formed during the labeling interval. Dairy products, calcium containing-binders and aluminium containing-antacids, interfere with the absorption and binding of tetracycline and should be avoided. Nevertheless, in case of failure or absence of tetracycline labeling, the evaluation of static histomorphometric parameters of bone remodeling (e.g., active osteoblasts and osteoclasts per bone, osteoid and eroded perimeters osteoid per bone, and share of fibrosis) may provide acceptable information for the diagnosis of turnover in ROD: indeed, in a prospective study of BB performed before and post-kidney transplantation (*n* = 205), all static parameters were significantly different among high and low turnover and demonstrated good discriminatory ability (AUCs 0.70–0.84), especially for high bone turnover [39].

Recommendations to prevent complications include not to perform biopsy, in hemodialysis patients, on the dialysis day to avoid hematoma or bleeding from heparin exposure; in peritoneal dialysis patients, not having a day dwell of dialysis fluid on the day of the biopsy because of the concomitant sedation.

BB should be performed in a procedural area that is equipped to monitor a sedated patient. Sedation with propofol, in addition to local anesthesia with xylocaina, markedly improves the perception of patients and reduces pain; sedation should be preferentially performed in the presence of an anesthetist, such as in a recovery room. The fragment from the anterior iliac crest is obtained by manual puncture with a pointed trocar with a trephine whose internal diameter is 7.5 mm [40]. The complete presence of two cortical areas and trabecular bone should be assessed.

In consideration of relative invasiveness and costs, the accuracy of ROD diagnosis has been evaluated on halved histological bone sections (3.5 mm diameter) to determine if it is comparable to the standard 7.5 mm findings. The final diagnosis was in full accordance between the whole biopsy and the two corresponding hemi-biopsies in 91% of cases [41]. Although these positive data could increase the use of biopsies in CKD, the use of smaller needles deserves attention: in that study, bone samples were originally obtained with larger trephines, expecting to produce lower artefacts compared with thinner needles. A smaller sample may result in loss of information, particularly in terms of turnover, but they could be useful as a follow-up tool to monitor the treatment of patients already diagnosed and characterized through a standard biopsy [42].

After the procedure, the patient should stay for 3–6 h to allow the effects of the sedation to wear off before going home the same day. Bone samples are fixed in ethanol 70%, followed by dehydration in 96–99.9% ethanol; it should be sent to dedicated hospitals or laboratories to further process and conduct analysis within 5–7 days [36]. The sample received is embedded in methyl-methacrylate resin, then cut with a microtome equipped with a carbide-tungsten blade to obtain 8-mm sections, which were stained with Goldner’s trichrome and mounted on microscope slides for histomorphometric evaluation [43].

## 7. Bone Biopsy Practice across Europe

In May 2015, the current BB practice patterns and attitudes across Europe were evaluated through an electronic survey sent by mail to all European members of the ERA-EDTA CKD-MBD Working Group (*n* = 230) with a response rate of 32% [32]; the main activity of respondents was clinical nephrology (about 50%). Half of the respondents reported having performed biopsies over the last 5 years, mainly for clinical purposes, but unfortunately, the total number of BB procedures per respondent in that period was very low, being <10. Regarding technical issues, the additional data showed that small (internal diameter <5 mm) trephine needles are the most popular (40%) and that procedures were mainly performed with local anesthesia in combination with light sedation (66.7%). As expected, histomorphometric analysis was mostly performed in external laboratories. To date, in accordance with the data collected, biopsies are currently performed only in specific cases and in a limited number of centers across Europe, despite it is well considered the diagnostic gold standard in CKD-MBD. The European survey has reflected the perception of BB as invasive, painful, laborious, even if clinically useful procedure. The most frequently detected limitations were laborious sampling procedure, excessive time and costs of analysis, and especially lack of expertise; moreover, 51% of respondents stated how procedural pain is a definite obstacle to the implementation of BB, clear statements about reimbursement are missing in several European countries and even more alarming is the vanishing reservoir of expertise in histomorphometry. This ongoing negative spiral may ultimately result in the complete disappearance of BB expertise [1].

The creation of the European Renal Osteodystrophy (EUROD) initiative, in 2016, may interrupt this spiral; a European collaborative network is needed to revitalize BB in clinical practice and to facilitate research on epidemiology, implications, and reversibility of ROD. Thus, the shared purposes of the EUROD initiative currently include: the spread of expertise through organization of training programs for clinicians and pathologists, to facilitate comparison of results by harmonization of BB procedure and histomorphometric parameters, to promote pan-European research in this field with the creation of an online shared repository of existing clinical material and identification of research questions (e.g., whether more frequent bone biopsies result in a decreased fracture burden), to gain knowledge in the field of ROD leading to a collaborative update of guidelines and patient management, and also to closely interact with similar initiatives elsewhere in the world. In this way, implementation of BB will improve CKD patients’ management and will widen the therapeutic horizon [2,6,32].

## 8. Bone Biopsy Practice across Italy

As for Europe, in Italy BB is rarely performed in CKD patients; limitations that hamper diffusion of this approach are the same previously described. The main hurdle is the lack of expertise in performing and histopathological reading, in addition to the cost, invasiveness and potential pain associated to the procedure.

Considering the actual limited use of BB, the Italian Society of Nephrology (SIN) and the Italian Scientific Society of Osteoporosis and Mineral Metabolism Diseases of the Skeleton (SIOMMMS) and have joined forces (Figure 4).

**Hub and Spoke Italian model of BB.** To approach the lack of widespread expertise, the multidisciplinary commission by SIN and SIOMMMS has proposed in Italy a Hub and Spoke model since 2015. The Italian network of renal osteodystrophy has been started with the identification of two pathology Hubs in Verona and Roma; these centers committed to process and analyze BB samples throughout the country and to provide histopathological diagnosis within 30 days from receival. In this way, biopsy technical issues and sample submission procedure have been standardized and the relative documentation is today available on SIN and SIOMMMS websites available to all physicians [44].

**Procedure adequate to reduce the pain of BB.** Nowadays procedural invasiveness and pain represent recurrent limitations for the widespread implementation of BB as a diagnostic tool across Italy and Europe. The protocol adopted by the Italian Hubs proposes the administration of moderate to deep sedation, in addition to local anesthesia, to reduce pain and ameliorate the perception of patients. Sedation could be done with midazolam or propofol, with the latter recommended for optimal pharmacokinetics properties and manageability; BB should be performed in a procedural area equipped to monitor vital signs with the availability of a nearby anesthetist to manage potential adverse events of sedation. The ‘target controlled infusion’ of hypnotic drugs for sedation has been also proposed to easily achieve and maintain the serum concentration without the risk of over dosage. Italian experience has resulted in excellent biopsy conditions, significant reduction of perceived pain and rapid recovery; after some hours of rest, patients could go home and resume their normal routine activities, making BB a day case procedure.

**Educational Training.** The Italian collaborative initiative is continuously pursuing the diffusion of BB: organization of training programs has been promoted by both societies in order to improve expertise in performing BB and a series of educational meetings on technique will then take place in 2022 across Italy.

**New code for BB in Italian Health System to obtain an adequate national reimbursement of BB.** Lack of adequate reimbursement is a significant obstacle to diffusion of BB in several countries. Refund of BB is not centralized in Italy and, thus, it is subordinated to regional differences, with an accurate and appropriate reimbursement only guaranteed by Region of Veneto.

The collaborative commission by SIN and SIOMMMS has approached this issue by submitting a request of adjustment and national homogenization of reimbursement to Italian Health Ministry. Furthermore, a national nomenclature of BB, as ‘histodynamic bone biopsy’, will facilitate a unique code of reimbursement.

## 9. Conclusions

BB remains the gold standard to characterize metabolic bone disorders in CKD patients. Meticulous definition of ROD types, through histomorphometric criteria and dynamic assessment after double tetracycline labeling, is key to guiding and supporting therapeutic choices, leading to a positive impact on the management and prognosis of uremic patients. Many hurdles are continuing to threaten the uptake of BB, defining the urgent need to provide solid solutions; with this position paper, the Italian collaborative network of experts aims to share experience, promote expertise, and implement proper solutions to spread BB in clinical practice in Italy, but potentially all over Europe. Indeed, the optimization of technique and the construction of Hub and Spoke model ameliorate the perception of this procedure, standardize results, reduce costs, and promote research, with the final aim of significantly optimizing CKD patient management and prognosis.

## Figures and Tables

**Figure 1 nutrients-14-01742-f001:**
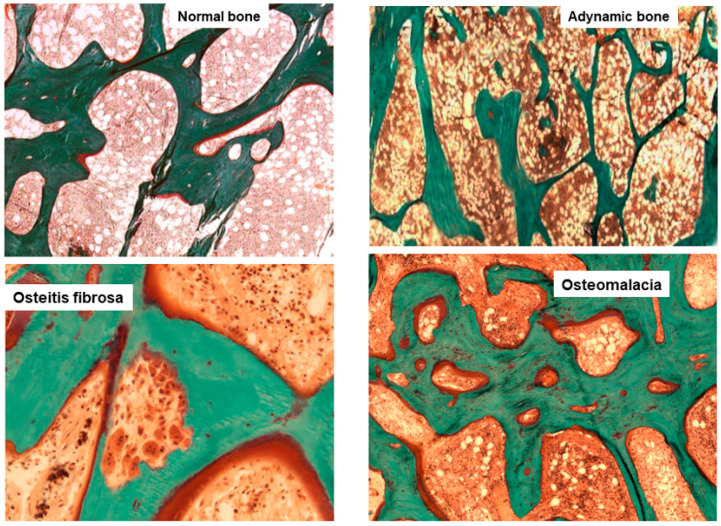
Main histomorphometric patterns of ROD in Goldner trichrome-stained bone section according to TMV classification system.

**Figure 2 nutrients-14-01742-f002:**
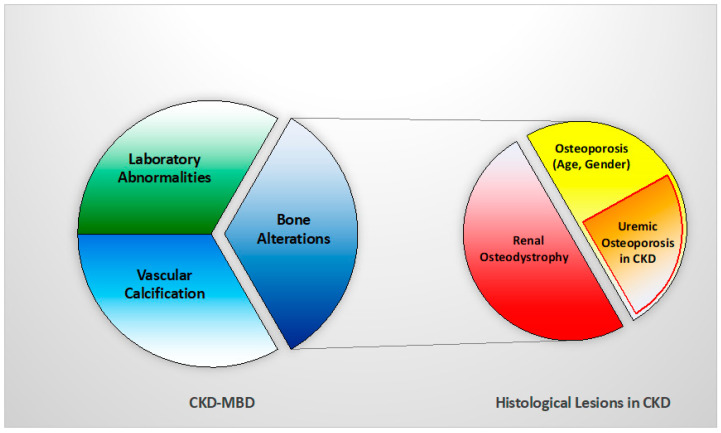
Relationship between CKD-MBD and bone histological patterns in CKD, like renal osteodystrophy and osteoporosis (associated with uremia and/or age-gender, among other factors). This relation impacts bone fragility and fractures susceptibility in CKD patients.

**Figure 3 nutrients-14-01742-f003:**
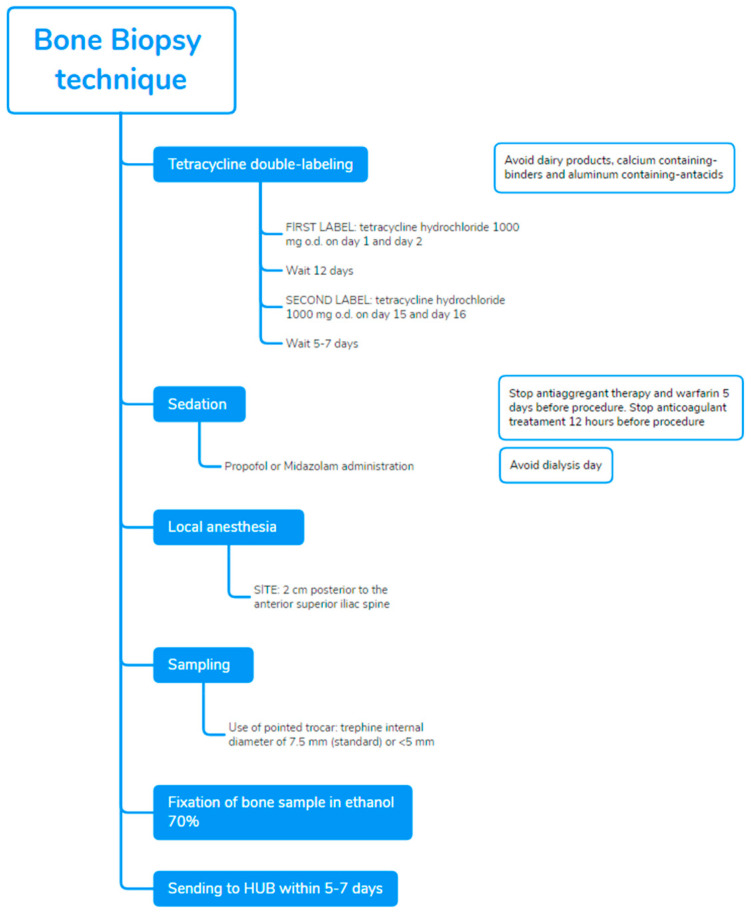
BB technique and tetracycline double-labeled procedure in accordance with the Italian proposal [36].

**Figure 4 nutrients-14-01742-f004:**
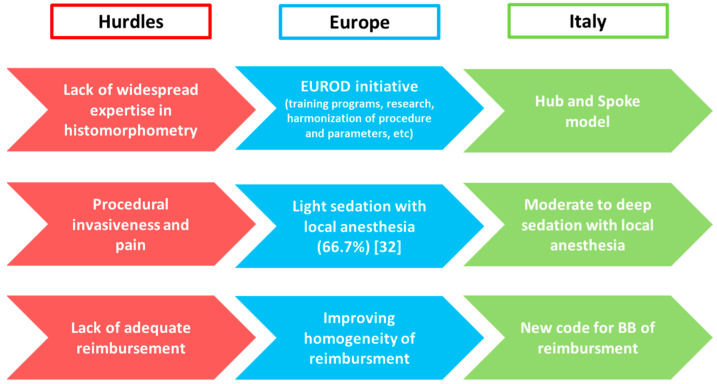
Main hurdles that hamper diffusion of BB across Europe and Italy and the specific solutions provided and implemented by Italian scientific societies (SIN and SIOMMMS), with the potential aim to also spread them to other countries [32].

**Table 1 nutrients-14-01742-t001:** Histologic classification of renal osteodystrophy in CKD with TMV classification.

Type of Renal Osteodystrophy	Turnover	Mineralization	Volume
Osteomalacia	Low	Abnormal	Low to Medium
Osteitis Fibrosa	High	Normal	Normal to High
Adinamic Bone Disease	Low	Normal	Low to Normal
Mixed Osteopathy	Normal to High	Abnormal	Low to Normal
Osteoporosis	Normal	Normal	Low

**Table 2 nutrients-14-01742-t002:** Comparison of main indications to perform BB in CKD patients, collected from the literature [14,32,33,34].

Evenpoel P et al. [32]	Brazilian Registry of Bone Biopsy (REBRABO) [14]	2009 KDIGO CKD-MBD Guideline [34]	2017 KDIGO CKD-MBD Guideline Update [33]
Low-impact fractures	Nontraumatic bone fractures	Multiple fractures	If knowledge of type of ROD will impact treatment decisions
Unexplained bone pain	Persistent bone pain	Persistent bone pain	
Prior to parathyroidectomy	Prior to parathyroidectomy	Suspected aluminium toxicity	
Prior to antiresorptive drugs	Prior to bisphosphonate therapy	Before osteoporosis treatment	
Unexplained hypercalcemia	Unexplained hypercalcemia/phosphoremia	Unexplained hypercalcemia	
Radiologic abnormality	Research protocol	Unexplained hypophosphatemia	
Suspected toxicity to heavy metals	Suspected aluminum accumulation		
Discordance between PTH and ALP levels

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
