# Peer review of "Time for Revival of Bone Biopsy with Histomorphometric Analysis in Chronic Kidney Disease (CKD): Moving from Skepticism to Pragmatism"

_nutrients, 2022, doi:10.3390/nu14091742_

Round 1

Reviewer 1 Report

This perspective paper evaluates the importance of bone biopsy in CKD and renal dystrophy.  The paper is well done.  It would be useful to include how a bone biopsy could reveal a treatment direction for the indications listed in Table 2 like unexplained hypercalcemia or hypophosphatemia.

Author Response

Response to Reviewers comments.

The Authors wish to thank the Editors and the three reviewers for their helpful and positive comments. All the revisions have been highlighted in red/blue in the manuscript and, consequently, we made the proper corrections of references in tables and bibliography.

Response: we would like to thank the reviewer for this good comment. Now, we have added more sentences (page 9, lines 414-426: BB currently remains the gold standard to confirm diagnosis of changes in mineral and bone metabolism and characterize different forms of ROD, providing most relevant information to guide choices in CKD-MBD management. Thus, the therapeutic approach in CKD-MBD is challenging and complicated by the fact that CKD patients may be affected both by aspects related to osteoporosis and by specific changes due to underlying forms of ROD. Performing BB before starting new treatment of CKD-MBD, allows to choose the best feasible strategies according to bone turnover: in the case of adynamic bone disease, the use of antiresorptive drugs (e.g. bisphosphonates, denosumab) is avoided preventing the excessive suppression of skeletal remodeling and/or precipitation of a condition of low turnover. Conversely, in the case of ROD with high turnover, the use of anabolic drugs (e.g. teriparatide, and not yet prescribed abaloparatide and romosozumab), is restricted limiting the significant increase in skeletal turnover. BB could be useful even as a follow up technique, able to verify the therapeutic effect of antiresorptive or anabolic drugs or their potential side effects on bone [37, 38]) explaining how bone histomorphometry could guide in practice therapeutical strategies in CKD-MBD (e.g. antiresorptive or anabolic drugs).

Cannata-Andía JB, Martín-Carro B, Martín-Vírgala J et al. Chronic Kidney Disease-Mineral and Bone Disorders: Pathogen-esis and Management. Calcif Tissue Int. 2021 Apr;108(4):410-422.

Giusti A, Fusaro M. Il trattamento del paziente fratturato con insufficienza renale cronica (CKD) [The treatment of the pa-tient presenting with chronic kidney disease (CKD) and fragility fractures]. G Ital Nefrol. 2017 Dec 5;34(Nov-Dec):2017-vol6. Italian.

Reviewer 2 Report

This multidisciplinary position paper discusses the importance of bone biopsy in assessing status of renal osteodystrophy and in achieving a correct diagnosis. It widely describes CKD-MBD with related diagnostic tools and includes bone biopsy indications with technique procedure suggestions. It represents a real support for clinicans involved in the management of renal osteodystrophy.

Author Response

Response to Reviewers comments.

The Authors wish to thank the Editors and the three reviewers for their helpful and positive comments. All the revisions have been highlighted in red/blue in the manuscript and, consequently, we made the proper corrections of references in tables and bibliography.

Response: we would really thank the reviewer for these good and positive remarks.

Reviewer 3 Report

Fusaro and colleague propose an article highlighting the importance and the need for revival for bone biopsies in chronic kidney disease. This paper is of importance and explain quite clearly the present situation and why the bone biopsy although invasive is compulsory to obtain a proper bone diagnosis. Although this paper reveals the Italian situation towards bone biopsies, it also shed some light on the European situation and suggest not only Italian solutions but solutions that could be transposable in any country, and could end up with new collaborative network in order to improve the chronic kidney disease patient care.

Few comments are however needed.

Some pictures to illustrate the Table 1 would be informative.

Throughout the manuscript mention is made about the expertise needed to correctly conduct the histomorphometry analyses of the bone biopsies. Although the tetracycline procedure, the biopsy procedure together with the adequate sedation protocol and the fixation procedure is clearly stated, in addition to what information is collected from the bone biopsies, no real explanation of how the bone biopsies are analyzed is given. Obviously, this might be out of the scope of this paper, but maybe few details could be added.

Apart from those comments, the paper is fit for publication.

Author Response

Response to Reviewers comments.

The Authors wish to thank the Editors and the three reviewers for their helpful and positive comments. All the revisions have been highlighted in red/blue in the manuscript and, consequently, we made the proper corrections of references in tables and bibliography.

Response: we thank the reviewer for his observation and we agree that some bone biopsy pictures can make the manuscript more attractive and informative to the readers. We have now added in the text some significant pictures of different patterns of ROD (Figure 1, page 4, lines 158-159: Figure.1 Main histomorphometric patterns of ROD in Goldner trichrome-stained bone section according to TMV classification system.) and a further reference, also by our working group, dealing with this special issue (page 3, lines 154-156: Suggestive representations of these different forms of ROD are obtained by BB approach, allowing the study of each component of bone multicellular unit (Figure.1) [5].). We have also added few sentences of sample analysis description after fixation procedure to give more exhaustive details on technique (page 12, lines 526-529: The sample received is embedded in methyl-methacrylate resin, then cut with a microtome equipped with a carbide-tungsten blade to obtain 8-microm sections, which were stained with Goldner’s trichrome and mounted on microscope slides for histomorphometric evaluation).

Dalle Carbonare L, Valenti MT, Giannini S, et al. Bone Biopsy for Histomorphometry in Chronic Kidney Disease (CKD): State-of-the-Art and New Perspectives. J Clin Med. 2021;10(19):4617.

Bedogni A, Saia G, Bettini G et al. Osteomalacia: the missing link in the pathogenesis of bisphosphonate-related osteone-crosis of the jaws? Oncologist. 2012;17(8):1114-9